# Oxidized Dietary Oil, High in Omega-3 and Omega-6 Polyunsaturated Fatty Acids, Induces Antioxidant Responses in a Human Intestinal HT29 Cell Line

**DOI:** 10.3390/nu14245341

**Published:** 2022-12-15

**Authors:** Tone-Kari Knutsdatter Østbye, John-Erik Haugen, Elin Merete Wetterhus, Silje Kristine Bergum, Astrid Nilsson

**Affiliations:** Nofima AS, Norwegian Institute of Food, Fisheries and Aquaculture Research, 1433 Ås, Norway

**Keywords:** lipid oxidation, hydroxyalkenals, dietary oxidized oil, intestinal cells, antioxidant defence

## Abstract

When oxidized, dietary oils generate products which have the potential to cause adverse effects on human health. The objective of the study was to investigate whether lipid oxidation products in an oxidized dietary oil can be taken up in intestinal cells, induce antioxidant stress responses and potentially be harmful. The in vitro cell model HT29 was exposed to camelina oil with different extents of oxidation, or only 4-hydroxy-2-hexenal (HHE) or 4-hydroxy-2-nonenal (HNE). The cellular content of HHE increased with an increasing extent of oxidation of the camelina oil added to the cell’s growth media, whereas HNE did not show a similar trend. Deuterated HHE was taken up by the HT29 cells, with 140 µM HHE metabolized within 0.5–1 h. The low oxidation degree of the camelina oil increased the gene expression of antioxidant markers (GPX, ATF6, XBP1). The increase in the gene expression of SOD at medium oxidation levels of the oil might indicate different regulation mechanisms. Highly oxidized camelina oil and a low concentration of HHE, over time, induced SOD and catalase enzyme activity in HT29 cells. Oxidized camelina oil contains multiple oxidation products which can be responsible for the intracellular responses observed in HT29 cells, while HHE and HNE in combination with other oxidation products induce antioxidant defence responses.

## 1. Introduction

Guidelines released by many medical organizations, including the World Health Organization (WHO) [1], have advocated for reducing the intake of saturated fat and increasing the intake of polyunsaturated fat, especially omega-3 polyunsaturated fatty acids (PUFAs), to promote health and reduce the risk from cardiovascular diseases. PUFAs, classified as omega-3 and omega-6 fatty acids, regulate a wide range of functions in the body, including blood pressure, blood clotting, and the correct development and functioning of the brain and nervous system [2]. Furthermore, PUFAs have a role in regulating inflammatory responses through the production of inflammatory mediators termed eicosanoids and resolvins [2,3,4]. The double bonds in PUFAs are highly available for lipid peroxidation. During the storage and processing of food products that include dietary oils, several volatile and non-volatile lipid oxidation products are produced, including possible genotoxic and cytotoxic compounds, such as 4-hydroxy-2-alkenals; at present, the dietary effect of such oxidized food is still unclear.

In the body, both enzymatic and non-enzymatic mechanisms can lead to lipid peroxidation. The enzymatic mechanisms involve lipoxygenases, cyclooxygenases, and cytochrome P450 (reviewed by Niki et al. [5]). Depending on the PUFA substrates, products with high stereo-specificity, such as eicosanoids and resolvins, are generated [5,6]. Non-enzymatic mechanisms are mediated by free radicals and result in a mixture of nonspecific stereoisomers, including hydroxyalkenals including 4-hydroxy-2-hexenal (HHE) and 4-hydroxy-2-nonenal (HNE) from omega-3 and omega-6 fatty acids, respectively [7]. Lipid peroxidation products generated by both mechanisms modulate a multitude of physiological processes, such as cell signalling, skin barrier function, coagulation, immune responses, and inflammation. Oxidation products from non-enzymatic lipid oxidation are considered to be more toxic compared to products from enzymatic lipid oxidation [8].

Oxidation is the major cause of the loss of quality in dietary oils and lipid foods because unsaturated fatty acids, especially PUFAs, are highly susceptible to peroxidation during the storage and processing of food products [9]. Peroxidation causes an unpleasant smell and/or taste, a loss of nutritional quality, and may further lead to the generation of possible genotoxic and cytotoxic compounds, such as 4-hydroxy-2-alkenals [7,10,11]. Evidence that HNE and HHE in oxidized food can be absorbed in the intestine has been demonstrated [12], but in our previous study no increases in either HNE or HHE in healthy humans were observed after intaking oxidized cod liver oil over a period of 7 weeks [13]. Many cell types, such as intestinal cells, have defence systems, of which the antioxidant enzymes GPX (glutathione peroxidase), SOD (superoxide dismutase), and catalase help to detoxify and degrade lipid oxidation products [14,15,16]. In addition, transcription factors, such as ATF6 and XBP1, are activated specifically in response to endoplasmic reticulum (ER) stress (reviewed by Reads and Schröder [17]).

In the past three decades, total fat and saturated fat intake as a percentage of total calories has continuously decreased in Western diets, the intake of omega-6 fatty acids has increased, and the intake of omega-3 fatty acids has decreased. This large shift in the omega-6/omega-3 ratio, from 1:1 to 20:1 during evolution, or even higher today, might be responsible for negative human health effects [18]. Most vegetable oils contain high amounts of the omega-6 fatty acid linoleic acid (LA, 18:2n-6) and lower levels of the omega-3 fatty acid α-linolenic acid (ALA, 18:3n-3). Camelina (*Camelina Sativa* L. Crantz) oil is an underexploited, but promising dietary oil with high contents of both omega-3 and omega-6 PUFAs. Camelina seeds contain up to 45% oil, which is particularly high in omega-3 fatty acid ALA, constituting 35–45% of the total fatty acids [19,20]. The content of the omega-6 fatty acid LA in camelina oil is about half of the ALA content. We have recently shown that plant rest materials with natural antioxidants can increase the oxidative stability of an omega-3 rich Norwegian cold-pressed *Camelina sativa* oil [21]; however, increased oxidation and elevated levels of 4-hydroxy-2-alkenals were observed during storage experiments.

While our understanding of in vivo lipid oxidation as it relates to health and diseases has increased [8,22], less is known about the health effects of consuming foods that contain oxidized lipids. The objective of the present study was to investigate if lipid oxidation products in an oxidized dietary oil can be taken up in intestinal cells, induce antioxidant stress responses, and potentially be harmful. The in vitro cell model HT29, which is a colon cancer cell line, and camelina oil at different oxidation levels were used in the study.

## 2. Materials and Methods

### 2.1. Dietary Oil with Different Degrees of Oxidation

Norwegian cold-pressed camelina oil (*Camelina sativas* L.) (CO) was purchased from Norsk Matraps AS (Tomter, Norway). The camelina oil contained only 10% (*w*/*w*) saturated fatty acids (SFA), 32% (*w*/*w*) monounsaturated fatty acids (MUFA), and 57% (*w*/*w*) polyunsaturated fatty acids (PUFAs) (Table 1). Linoleic acid (18:2 n-6) and α-linolenic acids (18:3n-3) were the main PUFAs with 16% (*w*/*w*) and 37% (*w*/*w*), respectively. Oil samples with different oxidation degrees for the in vitro cell trial were produced by incubation of oil samples in the dark at 40⁰C with access to oxygen for 2, 4, 6, and 9 weeks as described in Nilsson et al. [21]. Initial samples (CO-0w) and samples taken after 2 w, 4 w, 6 w, and 9 w (CO-2w, CO-4w, CO-6w, CO-9w) were flushed with nitrogen and frozen at –80 °C immediately after sampling. Lipid oxidation parameters of the dietary oil samples are presented in Table 2. Fatty acid composition in oils were determined according to Kirkhus et al. [23]. Peroxide value (PV), anisidin value (AV), and hydroxyalkenals (HHE, HNE) in oils are reported in Nilsson et al. [21].

### 2.2. Culturing of HT29 Cells

Three parallel cell batches of HT29 (human colon cell line, ATCC^®^ HTB38, Manassas, VA, USA) were cultured in growth media consisting of DMEM medium with 10% foetal calf serum (Gibco Life Technologies, Waltham, Massachusetts, USA), 1× NEAA (Gibco Life Technologies), and 100 U/mL Penicillin-Streptomycin. The cells were maintained at 37 °C with 5% CO_2_ in a humidified incubator. Six parallel samples were used for each treatment, with 2 parallels per cell batch. The cells were seeded out to a density of 1 × 10^5^ cells/T25 cell culture flask (25 cm^2^) and rested for 24 h before substrates were added. Cells were seeded out in cell flasks (25 cm^2^) for analysis of hydroxyalkenals and enzyme activity, in 6-well plates (9.6 cm^2^) for gene expression analysis, and in 96-well plates (0.3 cm^2^) for viability assay, with cell densities adjusted accordingly.

### 2.3. Preparation of Substrates for The In Vitro Studies in HT29 Cells

The camelina oil substrates for the cell trial were prepared by mixing 50 mg of camelina oil with 0.1 mL DMSO (Sigma-Aldrich, Darmstadt, Germany), followed by vortexing and sonication for 8 s at amplitude 3. The oil mixture was then transferred to 10 mL foetal calf serum, sonicated for 8 s to create small micelles to facilitate its uptake into the cells and finally diluted to 1:10 in growth media without serum. The final concentration of the camelina oil in growth media was 0.5 mg/mL (with 10% foetal calf serum).

HHE and HNE (Cayman Chemical Company, Ann Arbor, MI, USA) were firstly dissolved in ethanol and thereafter diluted in growth media to the final concentration used in the different trials. The final concentration of ethanol in the growth media was 0.7%. In the cell trial evaluating cellular responses of different concentrations of HNE and HHE on H729 cells, final growth media concentrations of 0.0, 2.2, 6.7, 20.2, or 60.6 nM HNE or 3.1, 9.2, 27.6, or 82.9 nM HHE were used. Both in the time response study of HHE and the verification trial, 0.014 µM and 140 µM HHE were used.

### 2.4. Cell Trials

Four different cells trials were conducted: Trial 1: Cellular responses to oxidized camelina oil (oxidized oil study); Trial 2: Cellular responses to two different concentrations of HHE and HNE (concentration study of HHE and HNE); Trial *3*: Cellular responses to HHE over time (time response study of HHE); and Trial 4: Verification of HHE uptake into HT29 cells (verification study).

Selection of concentration of oil and hydroxyalkenals (HHE, HNE) for the cell studies was based on previous in vitro studies on hydroxyalkenals [12,24,25,26,27]. In these studies, different in vitro cultured cells were exposed to a concentration of 1–100 µM of HNE or HHE. In our cell trials, we used both lower and higher doses of HNE and HHE, from 0.1 nM to 140 µM.

The cells were washed in serum-free medium and then incubated with the growth media supplemented with oxidized camelina oil substrates or hydroxyalkenals. In trial 1, HT29 cells were exposed to oxidized camelina oil (0.5 mg/mL, preparation of substrate as described above) and incubated for 72 h before harvesting cells and media for the different analyses. In trial 2, HT29 cells were exposed to 0.0, 2.2, 6.7, 20.2, or 60.6 nM HNE or 3.1, 9.2, 27.6, or 82.9 nM HHE and incubated for 48 h for analyses of viability, hydroxyalkenals, and enzymes, and for 6 h for analysis of gene expression. In trial 3, the time response study of HHE, the HT29 cells were incubated with 0.014 µM (low) and 140 µM HHE (high) for 0.2, 0.5, 1, 2, or 24 h. In trial 4, HT29 cells were exposed to 140 µM of deuterated HHE (D3-HHE) for 0.5 h. The control in the studies with hydroxyalkenals was ethanol (0.1%) added to a similar concentration as to the cells added to growth media with hydroxyalkenals.

Growth media for analysis of hydroxyalkenals were immediately frozen at −80 °C. The cells were harvested by washing them twice in PBS. Cells for gene expression analysis were harvested in buffer RLT (Qiagen, Valencia, CA, USA), to which DTT was added (Sigma-Aldrich, Darmstadt, Germany), homogenized with Qiashredder following the manufacturer’s protocol, and stored at −80 °C until isolation of RNA. Cells for both analyses of hydroxyalkenals and enzyme activity were harvested in PBS and centrifuged at 2000× *g* for 5 min. The supernatants were removed, and the cells were stored at −80 °C until analyses.

### 2.5. Hydroxyalkenals in Cells and Growth Medium

Cell pellets (cells sampled from T25 cell culture flask of 25 cm^2^) and samples of growth medium (5 mL for trials 1 and 3, 3 mL for trial 4), both frozen and stored at −80 °C, were added to 10 ng of the internal quantification standards D3-HHE and D3-HNE (Cayman Chemical Company, Ann Arbor, MI, USA) before extraction and analyses of hydroxyalkenals as described for oil (point 2.1).

A modified in-house validated method [21] based on Luo et al. [28] was used. Frozen stored oils (−20 °C) and cell pellets (cells sampled from T25 cell culture flask of 25 cm^2^) were added to 10 ng of the internal quantification standards) D3-HHE and D3-HNE (Cayman Chemical Company, Ann Arbor, MI, USA). Prior to GC/MS analysis, pentafluorobenzyl- oxime-trimethylsilyl ether (PFB-oxime-TMS ether) derivatives of the syn- and anti-stereoisomers of the respective 4-hydroxyalkenals were generated during a two-step derivatization. An Agilent 7890A gas chromatograph interfaced with a 5975C mass selective detector (Agilent Technologies, Little Falls, DE) was used with 1 μL splitless injection. The PFB-TMS derivatives were separated on a HP-5MS fused silica capillary column (30 m × 0.25 mm × 0.25 μm) using helium as carrier gas at a flow rate of 1 mL/min. The oven temperature was programmed from 50 °C (1 min) at 10 °C/min to 240 °C, followed by 20 °C/min to 300 °C (5 min). Transfer line temperature was maintained at 280 °C. Derivatized aldehydes were measured in negative ion chemical ionization (NCI) mode. Methane was used as reagent gas with source pressure 2.3 × 10^−4^ Torr. Ion source temperature was 230 °C, with electron ionization energy of 100 eV. Mass spectra of derivatized standard compounds of HHE and HNE were first recorded in full scan for identification of target ions then for quantitation in selected ion monitoring (SIM) mode. The two syn- and anti-isomers of their respective PFB-oxime-TMS ether derivatives were monitored at *m*/*z* 291 (HHE) and *m*/*z* 283 (HNE); quantification was performed by measuring *m*/*z* 294 and *m*/*z* 286, respectively, of the deuterated D3-HHE and D3-HNE internal standards. Repeatability of the analysis of the two 4-hydroxyalkenals measured in replicate digest samples was within 10%. The instrumental limit of detection was 9 pg for D3-HNE and 14 pg for D3-HHE (S/N = 3).

### 2.6. Gene Expression

Total RNA was isolated from cells using RNeasy Plus Mini Kit (Qiagen, Valencia, CA, USA) following the manufacturer’s protocol. Concentration and purity of RNA were measured with a NanoDrop ND-1000 Spectrophotometer (NanoDrop Technologies, Wilmington, DE, USA). cDNA was synthesized from 500 ng RNA in a 20 µL reaction using TaqMan^®^ Reverse Transcription Reagents (Applied Biosystems, Foster City, CA, USA). Random hexamers were used to prime the reaction. The cDNA synthesis was run under the following conditions: 25 °C for 10 min, 37 °C for 30 min, and 95 °C for 5 min. The qPCR reaction mixture consisted of 4 μL diluted (1:10) cDNA, 1 μL forward and reverse primer (final concentration of 0.5 μM, Appendix A), and 5 μL SYBR Green-I Master (Roche Applied Science, Germany), and the reaction conditions were 95 °C for 5 min, 45 cycles at 95 °C for 15 s, and 60 °C for 1 min. The qPCR reaction was run on a LightCycler 480 (Roche Diagnostics Gmbh, Mannheim, Germany). Analysis of the melting curve of each primer pair was carried out to confirm amplification of only one PCR product. GAPDH, RPOL2, and EF1A were evaluated as reference genes using RefFinder [29]. The relative gene expression level was calculated according to the ∆∆Ct method [30], using GAPDH as the reference gene, and calculation of gene expression in the various treatments was relative to that of the control treatment.

### 2.7. Viability of Cells

Viability of the cells was evaluated using CellTiter-Glo^®^ Luminescent Cell Viability Assay (Promega, Madison, WI, USA) quantifying the ATP present as an indicator of metabolically active cells. The viability of cells from all treatments was between 94 and 100%, indicating that the cells were not considerably negatively affected by the treatments.

### 2.8. Enzyme Activity

#### 2.8.1. Superoxide Dismutase

Enzyme activity of superoxide dismutase in the cells was measured with Superoxide Dismutase (SOD) Activity Assay Kit (BioVision, Waltham, MA, USA). The cell pellet was resuspended in PBS (150 µL) and diluted to 1:10 before being assayed according to the manufacturer’s protocol.

#### 2.8.2. Catalase

Enzyme activity of catalase was measured according to the method described in Kjær et al. [31]. The cell pellet was resuspended in PBS (150 µL) and diluted to 1:10 before being assayed according to the manufacturer’s protocol.

### 2.9. Statistical Analysis

Statistical analyses were evaluated using the software package UNISTAT (London, England). The cellular contents of HHE and HNE (n = 2 for the oxidation study, n = 4 for the verification study), gene expression (n = 4), and enzyme activity (n = 4) were subjected to multiple comparison using one-way ANOVA followed by Tukey–HSD test (*p* < 0.05). Statistical units are different cell batches.

## 3. Results

### 3.1. Hydroxyalkenals in Cells and Cellular Responses to Oxidized Camelina Oil (Oxidized oil Study; Trial 1)

The first cell trial was performed to investigate if oxidized dietary oils with defined levels of oxidation products from both omega-3 and omega-6 PUFAs are taken up by intestinal cells and induce intracellular antioxidant responses by analysing the hydroxyalkenals in the cells, gene expression, and enzyme activity of the antioxidant markers.

#### 3.1.1. Hydroxyalkenals in Cells

The cultivation of HT29 cells in the growth media and camelina oil showed increased cellular levels of HHE with increases in the extent of oil oxidation (Figure 1). The intracellular content of HHE was 1.5–2.6 times higher than that in the control cells with no added oil, varying from 2.4 to 4.1 ng HHE/cell flask. The highest concentration of HHE was detected in the cells cultured with the most oxidized camelina oil (CO-9w), showing 4.1 ng/cell flask. The control cells without camelina oil in the growth media had an intracellular content of HHE of 1.6 ng/cell flask.

There was no increase in the cellular content of HNE when increasing the medium content, as observed for HHE. CO-0w, CO-2w, and CO-9w all had significantly higher contents of HNE compared to those of the control and CO-4w. The cellular level of HNE in the control was 2.4 ng/cell flask, whereas the cellular content of HNE in the other groups ranged from 2.4 ng/cell flask (CO-4w) to 3.9 ng/cell flask (both CO-0w and CO9w).

Only one growth medium sample per group was analysed for HHE and HNE content. The media from the control cells that were not cultured in camelina oil had 1.7 ng/cell flask of HHE and 2.1 ng/cell flask of HNE. The media from the cells cultured with camelina oil had an increasing content of HHE: 3.1 (CO-0w), 3.4 (CO-2w), 3.9 (CO-4w), 4.1 (CO-6w), and 5.1 ng/cell flask (CO-9w). Compared to that of the control, the content of HHE was 1.8–3.0 times higher in the growth media and camelina oil (CO-2w, CO-4w, CO6w, CO-9w). The media content of HNE was lower than that for HHE, comparable with the oil concentrations of these compounds. There was no difference in the medium concentration of HNE between the control and the CO-0w (2.1 ng/cell flask). However, the addition of camelina oil with a higher oxidation level increased the growth media content of HNE: 2.4 (CO-2w), 2.6 (CO-4w), 2.9 (CO-6w), and 3.7 ng/cell flask (CO-9w).

The regression analysis showed a high correlation between the cellular level of HHE, the level of HHE in the oil (R^2^ = 0.93), and TOTOX of the oil (R^2^ = 0.93) added to the cells. There was no correlation between HNE content in the cells and HNE content in the growth media (*p* = 0.582).

#### 3.1.2. Gene Expression

Culturing the HT29 cells in growth media and camelina oil with increasing oxidation levels induced changes in the gene expression of several markers for antioxidant defence, ER stress, and inflammation (Figure 2 and Figure 3). Cells cultured in growth media and camelina oil with a low oxidation degree to the growth medium (CO-0w and CO-2w) showed increased gene expression of GPX2 compared to that of the control, whereas there were no differences in the gene expression level between the control and cells exposed to camelina oil with a higher degree of oxidation (CO-4w, CO-6w, and CO-9w). Only the most oxidized camelina oil (CO-9w) affected the gene expression of CAT in the HT29 cells, showing down-regulated gene expression compared to the control. The gene expression of SOD2 in the cells exposed to the three lowest oxidation levels of the oils (CO-0w, CO-2w, and CO-4w) were not different from the control; however, the gene expression in CO-6w was significantly higher than that of the control. From the high expression of SOD2 in the CO-6w, the amount dropped to the level in the cells exposed to the most oxidized oil (CO-9w). Camelina oil with low oxidation levels induced the up-regulation of markers for ER stress, both ATF6 and XBP1, in HT29 cells (CO-0w and CO-2w) compared to the control. Cells exposed to a higher oxidation level of the oil (CO-4w, CO-6w and CO-9w), however, showed lower gene expression of the ER stress markers than in the cells cultured in oil with lower oxidation levels (CO-0w and CO-2w). The gene expression of XBP1 in CO-9w was significantly lower than in the control.

#### 3.1.3. Antioxidant Enzymes

The cells exposed to the most oxidized camelina oil (CO-9w) showed increased enzyme activity of superoxide dismutase compared to that of the control and CO-2w (Figure 4). The activity was 22.3% higher in CO-9w compared to the control. The CO-9w also showed significantly higher catalase activity, compared to all the other groups, with 26.9% higher activity compared to that of the control (Figure 5).

### 3.2. Cellular Responses to Two Different Concentrations of HHE and HNE (Concentration Study; Trial 2)

The second cell trial was performed to investigate if the two lipid oxidation products, HHE and HNE, can explain the cellular responses to oxidized camelina oil observed in the first cell trial by analysing the gene expression and enzyme activity of the antioxidant markers.

#### 3.2.1. Gene Expression

Culturing HT29 cells for 6 h in growth media with increasing concentrations of both HHE and HNE induced minor changes in the gene expression of markers for antioxidant response and ER stress (Appendix A). The gene expression of SOD2 was lower in the cells cultured in the highest concentration of HNE compared to the other groups. The cells cultured in growth media with the highest doses of HHE (89.2 nM) and HNE (60.6 nM) also showed the lowest expression level of ATF6 but were only significantly different from the cells cultured with growth media added to 9.2 nM HHE and 20.2 HNE, respectively. There were no significant differences in the gene expression of SOD1, SOD3, GPX, XBP1, and CAT.

#### 3.2.2. Activity of Antioxidant Enzymes

There were differences in the activity of superoxide dismutase between the cells cultured in growth media with different concentrations of HHE and HNE for 48 h (Appendix A); however, no clear dose–response was observed. The cells given the highest dose of HHE showed higher activity than those given 3.1 nM or 27.6 nM HHE. The activity in the control was, however, not significantly different from the activity in the cells cultured in the three highest doses of HHE. There were no differences in the activity of superoxide dismutase between the cells cultured with different concentrations of HNE in growth media, while the activities were not significantly different from those of the control.

### 3.3. Cellular Responses to HHE over Time (Time Response Study of HHE; Trial 3)

The third cell trial was performed to investigate how fast HHE is taken up and metabolized in the intestinal cells by analysing the temporal intracellular content of HHE. In addition, the temporal effects of HHE on the antioxidant stress responses were studied by analysing the gene expression and enzyme activity of the antioxidant markers.

#### 3.3.1. Uptake of Hydroxyalkenals

Exposing HT29 cells to low (0.014 µM) and high (140 µM) concentrations of HHE resulted in temporal significant differences in the intracellular levels of HHE ranging from 3.2 ng/cell flask in the High/24hours group to 15.3 ng/cell flask in the High/0.5hours group (Figure 6). The cells cultured in the high concentration of HHE showed an increasing trend from 0.125 h to 0.5 h after the addition to the growth media. Thereafter, the intracellular concentration of HHE dropped significantly up to 24 h. The incubation time of 0.5 h was therefore chosen for the verification study. Culturing the cells in the low HHE concentration did not result in detectable differences in the cellular concentrations of HHE over time. However, significant differences between the Low and High groups were seen at the time points 0.125 h, 0.25 h, and 2 h. The High groups had intracellular concentrations of HHE 1.3–3.7 times higher, although a concentration was added to the growth media that was 10,000 times higher.

#### 3.3.2. Gene Expression

The highest dose of HHE (140 µM) induced increasing expressions of both SOD1 and SOD2 during the 24 h time course (Figure 7). Compared to the control and earlier time points, the gene expression of SOD1 was significantly higher after 2 h of incubation and that of SOD2 after 1 h incubation when cultured in 140 µM HHE. For both genes, the Low and High groups were significantly different at only 2 and 24 h. There were no differences in the gene expression of SOD1 and SOD2 between the different time points when cultured in the low concentration of HHE. The gene expression of XBP1 was, however, not significantly different over 24 h in the cells incubated with the highest concentration of HHE, while the expression in cells cultured in the low HHE concentration was significantly higher at 24 h than at 2 h.

#### 3.3.3. Activity of Antioxidant Enzymes

• Catalase

The activity of catalase decreased over time when the cells were cultured with 140 µM of HHE (Figure 8). At 24 h, the activity in the cells was 0.3 times the activity in the control. Cells cultured in 0.014 µM HHE, on the other hand, had the highest activity of catalase at 24 h compared to 0.2 h. Despite the lower concentration of HHE, the activity of catalase at 2 h and 24 h was significantly higher in the cells cultured in 0.014 µM versus 140 µM.

• GPX

The activity of GPX showed a similar pattern for the high and low concentrations of HHE as for the catalase (Figure 9). The activity in the cells cultured in 140 µM HHE decreased relative to the incubation time, with the lowest activity at 24 h. The cells cultured in 0.014 µM showed significantly higher activity at 24 h than at 0.5 h. Although cultured in a higher concentration of HHE, the cells showed lower activity of GPX at 1 and 2 h.

• SOD

The activity of SOD did not show a dose–response pattern as for GPX and catalase (Figure 10). Similarly to that of the other antioxidant enzymes, the activity of SOD was higher in cells cultured with 0.014 µM compared to 140 µM at 0.2, 1, and 24 h.

### 3.4. Verification of HHE uptake into HT29 cells (Verification study; Trial 4)

The fourth cell trial was performed to verify if lipid oxidation products, represented by HHE, are taken up by intestinal cells by analysing the intracellular content of deuterated HHE.

#### Cellular Content of HHE

The NCI-GC/MS analysis results of the PFB-silyl-ether derivatives of HHE in extracts from cells cultured in media with added deuterated HHE and the control without added deuterated HHE in comparison with a deuterated HHE quantification standard are shown in Figure 11. Deuterated HHE was only detected in cells cultured in growth media with 140 µM deuterated HHE for 0.5 h. Deuterated HHE in the cells were at low levels, i.e., 0.07 ± 0.006 µg/cell flask.

## 4. Discussion

It is well known that lipid oxidation is the major cause of the loss of quality in dietary oils and lipid-containing foods, with an increased frequency of unpleasant smells and tastes [9]. While the understanding of in vivo lipid oxidation has increased as it relates to health and diseases [8,22], less is known about the health effects of consuming foods that contain oxidized lipids [32]. In this study we show that an oxidized dietary oil high in both omega-3 and omega-6 PUFAs can induce changes in the antioxidant defence and ER stress of intestinal cells. Furthermore, the evidence for HHE and HNE in combination with other oxidation products as possible inducers for antioxidant defence is strengthened.

Oxidized dietary oils contain multiple oxidation products which can be responsible for intracellular stress responses, including the increased expression of the transcription factors involved in ER stress and the increased activity of antioxidant enzymes SOD, GPX, and catalase [11,14,15,16].

In our study, the increased gene expression of GPX2, ATF6, and XBP1 induced by camelina oil with low oxidation degrees (CO-0w, CO-2w) (Figure 2 and Figure 3) shows that these antioxidant markers are sensitive to low levels of lipid oxidation products. An increased expression of transcription factors, such as ATF6 and XBP1, might indicate an activated ER stress pathway, but to confirm this, additional markers (such as ERDJ4, CHOP Daverkausen-Fischer) should be analysed. However, the increase in the gene expression of SOD caused by camelina oil was only observed with a medium level of oxidized oil (CO-6w), which might indicate different regulation mechanisms of the antioxidant genes. This can partly be explained by the function of the antioxidant enzymes, in which SOD catalyses the dismutation of the superoxide anion into O_2_ and hydrogen peroxide (H_2_O_2_), whereas H_2_O_2_ is reduced to H_2_O by GPX in the cytosol, or by catalase (CAT) in the peroxisomes [33]. The underlying mechanism for gene regulation might be through the nuclear factor erythroid 2-related factor 2 (Nrf2), which has been shown to be involved in transcription regulation of antioxidant genes [34]. Lipid oxidation products from omega-3 and omega-6 fatty acids, such as HHE and HNE, respectively, are shown to regulate Nrf2 ([35]). In the study by Zhang et al. [35], HHE generated from omega-3 PUFAs was a more potent NRF2 inducer than 4-HNE derived from n-6 PUFAs. The differential regulation of the antioxidant enzymes has also been found earlier in tracheobronchial epithelial cells [36]. In our study, camelina oil with the highest extent of oxidation did not further increase the gene expression of the stress markers in the cells compared to that with the lower oxidation levels, but it reduced the gene expression to a level similar or even lower than that in the control group. This is in accordance with studies on epithelial cells exposed to a high concentration of stressors, which also lead to a down-regulation of antioxidant genes [36]. Gene expression analysis is a sensitive method that gives a snapshot of the transcript level at a specific time point and therefore might be different from the actual enzyme activity. Thus, the responses in the gene expression do not necessarily reflect the enzyme activity or protein expression. This may explain the results in our study in which the highest enzyme activities of SOD and catalase were detected in the group exposed to camelina oil with the highest degree of oxidation, in contrast to the gene expression levels for most of the genes. Omega-3 and omega-6 fatty acids, such as those found in camelina oil, can activate the peroxisome proliferator-activated receptors (PPARs) functioning as ligand-dependent transcription factors, regulating multiple metabolic and cellular processes, including antioxidant responses [37].

The oxidized camelina oil used in this study is a highly unsaturated dietary oil containing relatively high amounts of the omega-6 fatty acid LA and the omega-3 fatty acid ALA, and due to the high extent of unsaturation, these PUFAs are prone to oxidation. All oxidized oil samples had higher PV levels than those recommended by the Global Organization for EPA and DHA (GOED) (max levels of PV at 5 meq/kg) [38]. The AV values were below the GOED recommended value of 20 for all the oil samples, whereas the TOTOX was above 26 in the more oxidized samples. Oxidized camelina oil contains several volatile and non-volatile lipid oxidation products, as presented in Nilsson et al. [21], including HHE and HNE. The level of HHE in the cells was enhanced when the cells were cultured in the growth media and oxidized oil. The HHE and HNE generated from the omega-3 and omega-6 PUFAs have been shown to induce oxidative stress in other studies. Awada et al. [12] reported that administration of oxidized tuna oil to mice led to an accumulation of HHE over time in their plasma, and oxidative responses in plasma and mucosa were also observed. On the contrary, Tullberg et al. [39] concluded that human digests of their investigated marine oils and their content of HHE did not cause stress responses in human intestinal Caco-2 cells. In our study, the proportion of omega-3 and omega-6 fatty acids in camelina oil (omega-3/omega-6 = 2.3) was reflected by higher levels of HHE than HNE in the oxidized oil samples, and the measured content of HHE was 1.8–3.0 times higher in the growth media containing oxidized camelina oil compared to the control.

The regression analysis showed a high correlation between the cellular level of HHE and the level of HHE in the oil (R^2^ = 0.93), but no correlation between HNE in the cells and the HNE content in the growth media (*p* = 0.582) was observed. Thus, either HNE is not taken up similarly to HHE, or the HNE is metabolised faster than the HHE. Earlier studies showed that 0.1 mM HNE was completely metabolized within 3–6 min in hepatocytes [24,25], whereas 30 min was needed to metabolise 95% of 5 µM HNE in endothelial cells [26]. The HT29 cells in our study were exposed to the growth media with oxidized oil delivered as micelles for 72 h, and even though the cells may use a longer time to take up oxidation products from micelles compared to single oxidation products, the metabolic differences in HNE might explain the cellular results. Hydroxyalkenals, such as HHE and HNE, are known to be able to form protein adducts [40], and it has been suggested that HNE has a higher ability to form covalent protein adducts than HHE [41]. This could also explain why an intracellular increase in HNE was not observed as was the case for HHE. In our timed study, the incubation of HT29 cells with 140 µM (High group) of HHE showed significantly increased intracellular HHE after 30 min, and thereafter the level decreased. The High groups had an intracellular concentration of HHE 1.3–3.7 times higher than the Low groups, although the concentration added to the growth media was 10,000 times higher. The significantly decreased levels of HHE after 2 h suggest that the cells can handle the elevated HHE levels. These findings are comparable to those of the studies mentioned with HNE in hepatocytes and endothelial cells [26,42].

In our study, the concentration of HHE available through the oxidized camelina oil (0.1–10.1 nM) is comparable to the lowest concentration used in the timed study with pure HHE (14 nM). The gene expression of SOD1 and SOD2 was not significantly affected in the cells incubated with this low concentration of pure HHE. However, a similar concentration of HHE available through oxidized oil induced the upregulation of SOD2. This can be explained by other components in the oxidized oil also inducing antioxidant responses [11,43,44] or differences in the incubation time and matrix. In contrast, the high concentration of HHE (140 µM) showed an increased expression of the SOD genes over time supported by a trend towards the increasing activity of the antioxidant enzymes. This can indicate an increasing oxidative stress status in the cells over time. In contrast to our study, the exposure of Caco2 cells to 1.4 or 90 µM HHE did not affect cellular SOD2 protein levels [39]. The differences could be explained by the difference in the concentration of HHE in addition to cell types.

The increasing HHE levels observed in the cells incubated with oxidized oil may be explained in two possible ways. First, the growth medium with significant levels of secondary lipid oxidation products, including hydroxyalkenals (among others), may lead to the transport of these products across the cell membrane and into the cells. Another explanation may be that the high levels of reactive oxygen species in the growth medium may induce oxidative stress responses in the cells and promote further lipid oxidation, leading to intracellular formation and the accumulation of HHE [43]. To verify whether the oxidation products can be taken up by the intestinal cells, an in vitro cell study with the same HT29 cells was carried out using deuterated HHE. The fact that deuterated HHE could be measured in the cells, even though at low levels (0.07 ± 0.006 µg/cell flask) compared to what was added to the medium (49 µg/cell flask), demonstrates that HHE may pass through the cell membrane and into the cells. The far lower levels of deuterated HHE found in the cells compared to the growth medium levels may suggest that a major proportion of the intracellular HHE is derived from intracellular oxidation or metabolism.

## 5. Conclusions

In conclusion, the increased gene expression of GPX2, ATF6, and XBP1 by low oxidation levels of camelina oil shows that these antioxidant markers are sensitive to low levels of unstable oxidative molecules. The increase in the gene expression of SOD only at a medium oxidation level of the camelina oil might indicate different regulation mechanisms of the antioxidant genes. Highly oxidized camelina oil and a low concentration of HHE over time induces SOD and catalase enzyme activity in HT29 cells. Deuterated HHE is taken up by HT29; however, the cells apparently metabolize 140 µM HHE within 0.5–1 h. Oxidized camelina oil contains multiple oxidation products which can be responsible for these intracellular responses and the evidence for HHE and HNE in combination with other oxidation products as possible inducers for antioxidant defence is strengthened.

## Figures and Tables

**Figure 1 nutrients-14-05341-f001:**
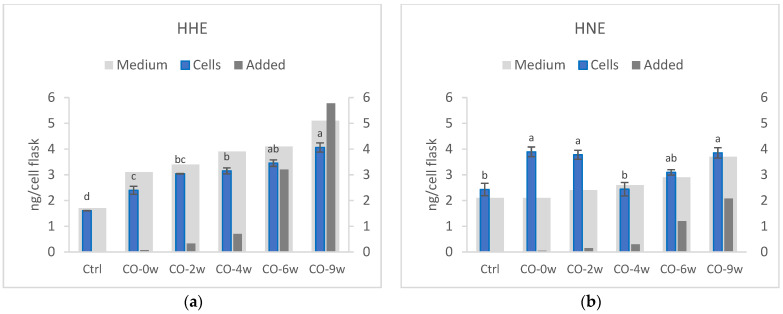
Cellular content of HHE (**a**) and HNE (**b**) (ng/T25 cell flask). The HT29 cells were cultured in growth media (Ctrl) or growth media and camelina oil (CO, 0.5 mg/mL) with increasing oxidation levels for 72 h. Cellular data (cells) are means (n = 2) shown with standard error mean. The figures also show the analysed medium concentration (Medium) of HHE and HNE as well as the calculated levels (Added) of HHE and HNE from the camelina oil added to the medium. Significant differences in cellular content of hydroxyalkenals are indicated with different letters evaluated with Tukey–HSD (*p* < 0.05). w = weeks.

**Figure 2 nutrients-14-05341-f002:**
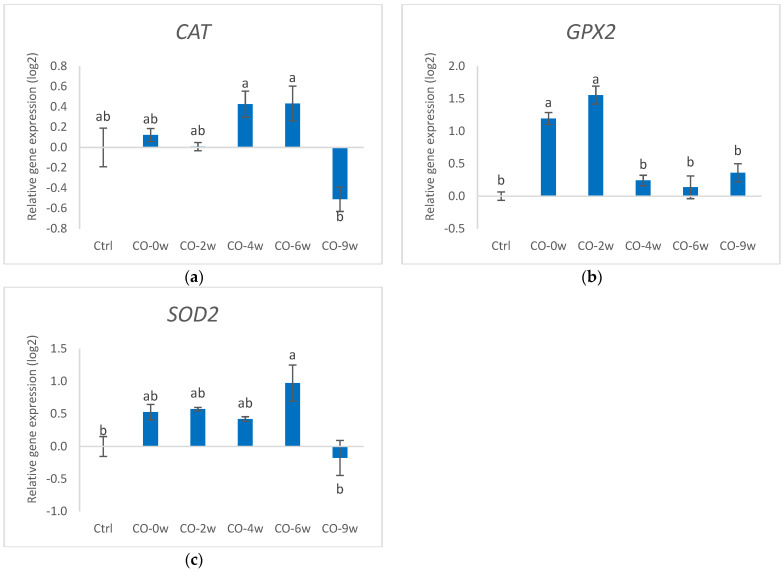
Relative gene expression (log) of markers for antioxidant defence: (**a**) CAT, (**b**) GPX2, and (**c**) SOD2 in HT29 cells exposed to oxidized camelina oil. All groups, except the control group (Ctrl), were cultured in growth media containing 0.5 mg/mL oxidized camelina oil for 72 h. Data (n = 4) are means shown with standard error mean. Significant differences are indicated with different letters evaluated with Tukey–HSD (*p* < 0.05). CO = camelina oil; w = weeks.

**Figure 3 nutrients-14-05341-f003:**
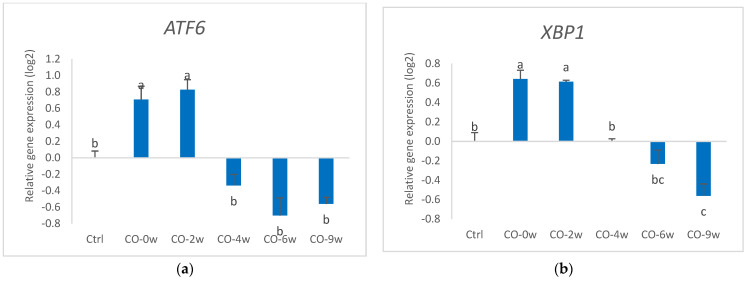
Relative gene expression (log) of markers for antioxidant defence: (**a**) ATF6 and (**b**) XBP1 in HT29 cells exposed to oxidized camelina oil. All groups, except the control group (Ctrl), were cultured in growth media containing 0.5 mg/mL oxidized camelina oil for 72 h. Data (n = 4) are means shown with standard error mean. Significant differences are indicated with different letters evaluated with Tukey–HSD (*p* < 0.05). CO = camelina oil; w = weeks.

**Figure 4 nutrients-14-05341-f004:**
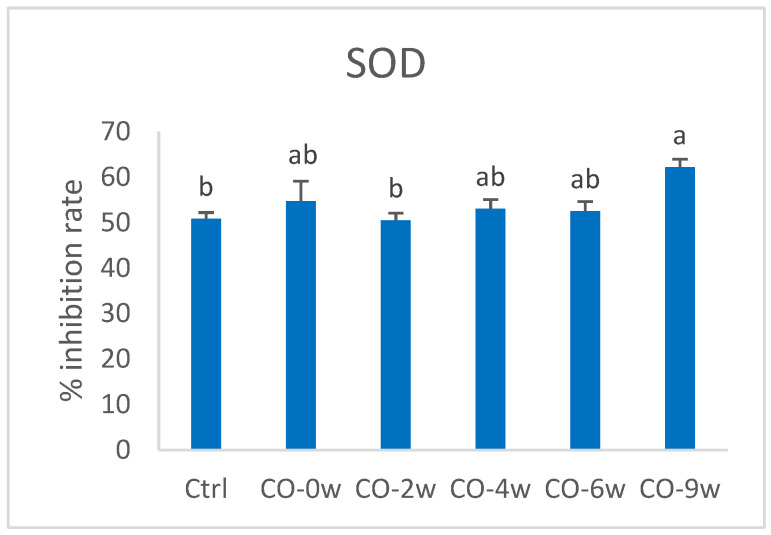
Activity of SOD (% inhibition rate) in HT29 cells exposed to oxidized camelina oil. All groups, except the control group (Ctrl), were cultured in growth media containing 0.5 mg/mL oxidized camelina oil for 72 h. Data (n = 4) are means shown with standard error mean. Significant differences are indicated with different letters evaluated with Tukey–HSD (*p* < 0.05). CO = camelina oil; w = weeks.

**Figure 5 nutrients-14-05341-f005:**
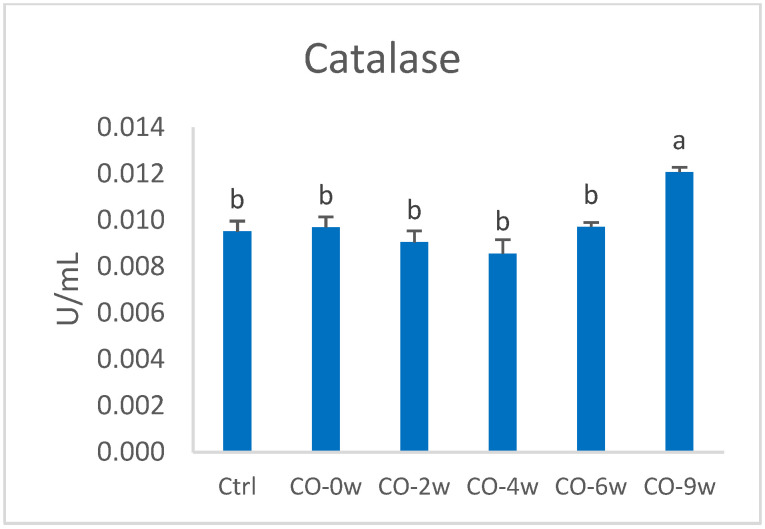
Catalase activity (U/mL) in HT29 cells exposed to oxidized camelina oil. All groups, except the control group (Ctrl), were cultured in growth media containing 0.5 mg/mL oxidized camelina oil for 72 h. Data (n = 4) are means shown with standard error mean. Significant differences are indicated with different letters evaluated with Tukey–HSD (*p* < 0.05). CO = camelina oil; w = weeks.

**Figure 6 nutrients-14-05341-f006:**
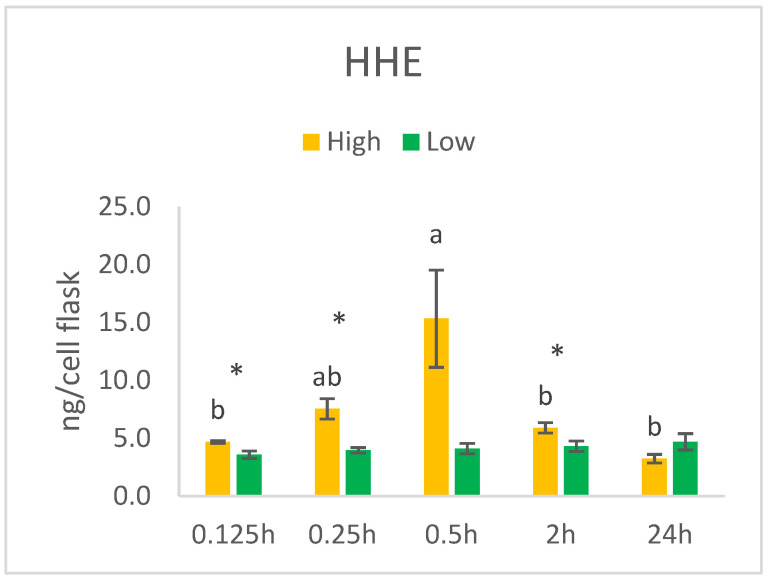
HHE (ng/cell pellet) in HT29 cells exposed to high (140 µM) or low (0.014 µM) concentrations of HHE. Data (n = 4) are means shown with standard error mean. Significant differences within Low or High groups are indicated with different letters evaluated with Tuke–-HSD (*p* < 0.05). Significant differences (*p* < 0.05) between Low and High evaluated by Student’s *t*-test are indicated with asterisk (*) at specific time points.

**Figure 7 nutrients-14-05341-f007:**
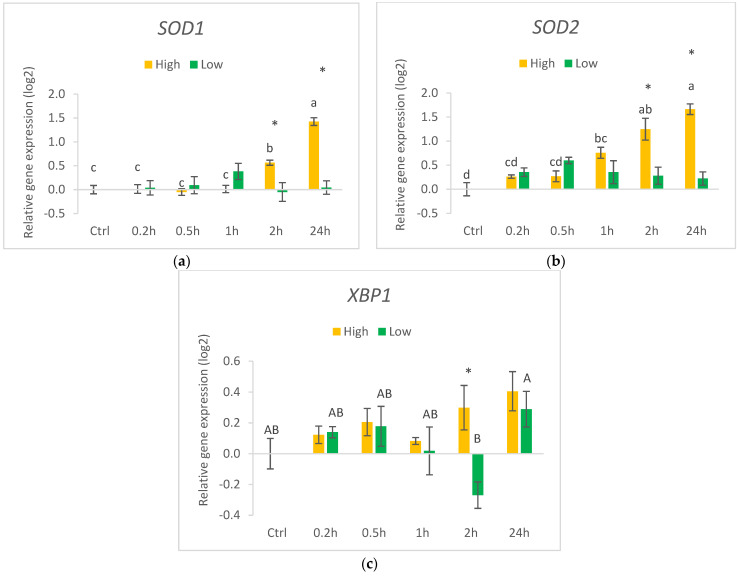
Relative gene expression of markers for antioxidant defence (**a**) SOD1, (**b**) SOD2 and ER stress (**c**) XBP1 in HT29 cells exposed to high (140 µM) or low (0.014 µM) concentrations of HHE in growth media. Data (n = 4) are means shown with standard error mean. Significant differences between control (Ctrl) and different time points within Low or High are indicated with different letters (High: lower letters, Low: capitalised letters) evaluated with Tukey–HSD (*p* < 0.05). Significant differences between Low and High are evaluated by Student’s *t*-test and indicated with asterisk (*) at specific time points.

**Figure 8 nutrients-14-05341-f008:**
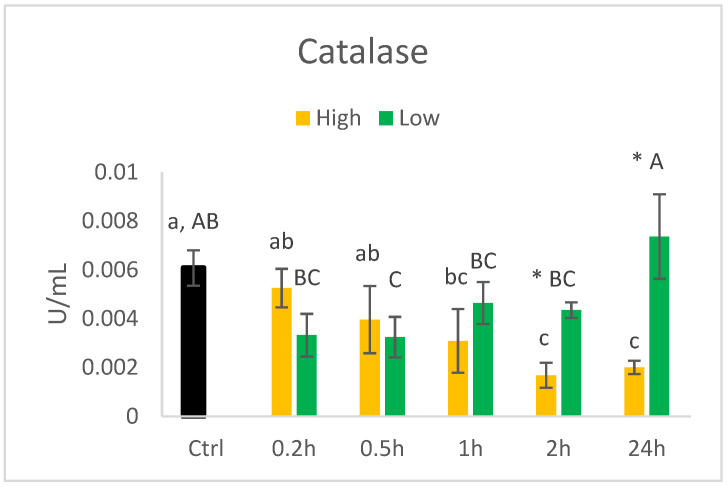
Activity (U/mL) of catalase in HT29 cells exposed to high (140 uM) or low (0.014 µM) concentrations of HHE. Data (n = 4) are means shown with standard error mean. Significant differences between control (Ctrl) and different time points within Low or High groups are indicated with different letters (High: lower letters, Low: capitalised letters) evaluated with Tukey–HSD (*p* < 0.05). Significant differences between Low and High are evaluated by Student’s *t*-test and indicated with asterisk (*) at specific time points.

**Figure 9 nutrients-14-05341-f009:**
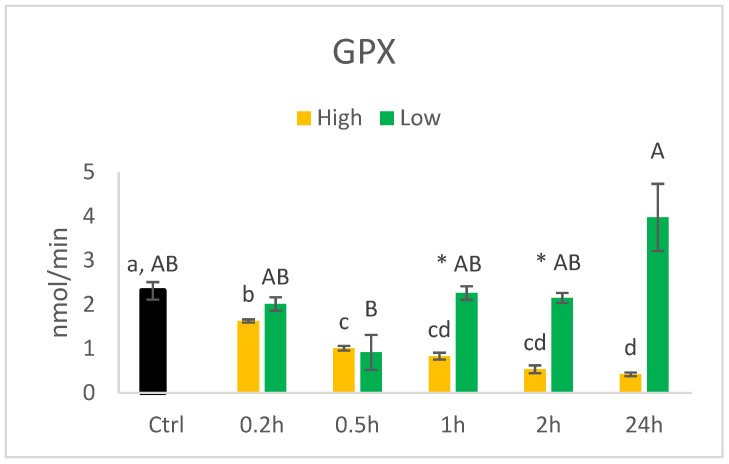
Activity (nmol/min) of GPX in HT29 cells exposed to high (140 µM) or low (0.014 µM) concentrations of HHE. Data (n = 4) are means shown with standard error mean. Significant differences between control (Ctrl) and different time points within Low or High groups are indicated with different letters (High: lower letters, Low: capitalised letters) evaluated with Tukey–HSD (*p* < 0.05). Significant differences between Low and High are evaluated by Student’s *t*-test and indicated with asterisk (*) at specific time points.

**Figure 10 nutrients-14-05341-f010:**
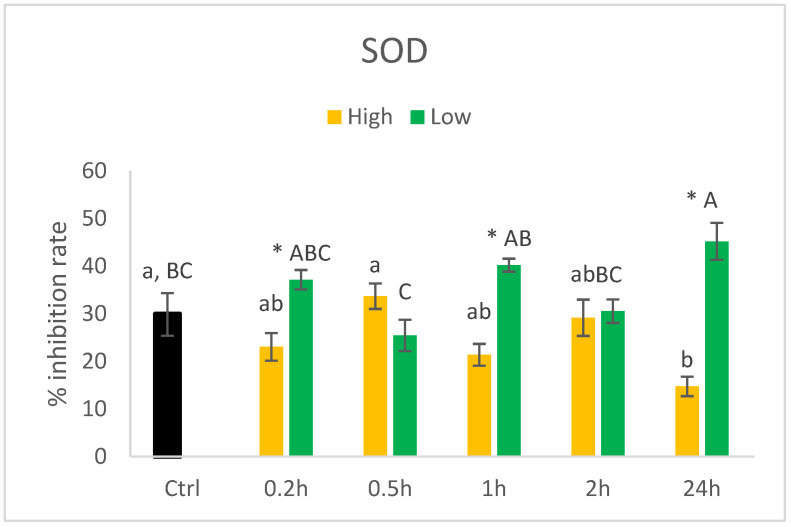
Activity (%inhibition rate) of SOD in HT29 cells exposed to high (140 µM) or low (0.014 µM) concentrations of HHE. Data (n = 4) are means shown with standard error mean. Significant differences between control (Ctrl) and different time points within Low or High groups are indicated with different letters (High: lower letters, Low: capitalised letters) evaluated with Tukey–HSD (*p* < 0.05). Significant differences between Low and High are evaluated by Student’s *t*-test and indicated with asterisk (*) at specific time points.

**Figure 11 nutrients-14-05341-f011:**
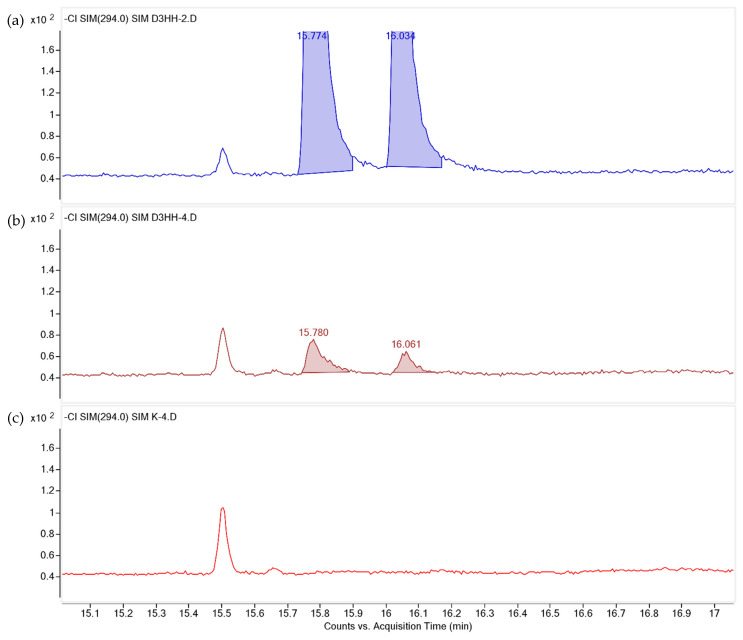
Identification of SIM (*m*/*z* 294) traces of syn- (15.774 min) and anti- (16.061 min) isomers of PFB-silyl-ether derivatives of deuterated HHE. The HT29 cells were cultured in growth medium (Ctrl) without HHE and growth medium with 40 µM deuterated HHE (n = 4) for 0.5 h. (**a**) Deuterated quantification HHE standard; (**b**) cells cultured in 140 µM deuterated HHE; and (**c**) cells cultured in growth medium without HHE.

**Table 1 nutrients-14-05341-t001:** Fatty acid composition of camelina oil (% of total fatty acids).

Fatty Acid % (*w*/*w*)	Camelina Oil
C14:0	≤0.5
C14:0	≤0.5
C16:0	5
C18:0	3
C20:0	2
C22:0	≤0.5
Sum SFA	10
C16:1	≤0.5
C18:1 n-9	13
C20:1 n-9	16
C22:1	3
C24:1	≤0.5
Sum MUFA	32
C18:2 n-6	16
C20:2	2
Sum n-6	18
C18:3 n-3	37
C20:3	2
Sum n-3	39

**Table 2 nutrients-14-05341-t002:** AV, PV (meq/kg), total oxidation value (TOTOX) (AV + 2×PV), and hydroxyalkenals (HHE and HNE, µg/g) of camelina oil (CO) stored in the dark at 40⁰C with access of oxygen up to 9 weeks (w). Data are mean (*n* = 2) shown with standard deviation. .

Sample	Time point (w)	AV	PV (meq/kg)	TOTOX	HHE (ug/g)	HNE (ug/g)
CO-0w	0	1.0 ± 0.0	5.9 ± 0.1	12.7 ± 0.1	0.03 ± 0.00	0.02 ± 0.00
CO-1w	1	1.0 ± 0.0	6.7 ± 0.1	14.3 ± 0.1	0.05 ± 0.00	0.03 ± 0.00
CO-2w	2	1.3 ± 0.0	11.0 ± 0.7	23.3 ± 1.4	0.13 ± 0.00	0.06 ± 0.00
CO-3w	3	2.5 ± 0.2	18.0 ± 0.9	38.4 ± 2.1	0.28 ± 0.04	0.11 ± 0.01
CO-4w	4	2.7 ± 0.2	15.3 ± 1.4	33.3 ± 3.0	0.35 ± 0.01	0.12 ± 0.00
CO-6w	6	8.0 ± 0.0	32.9 ± 0.5	73.7 ± 1.0	1.28 ± 0.11	0.48 ± 0.03
CO-9w	9	11.4 ± 0.1	40.3 ± 0.5	91.9 ± 1.1	2.31 ± 0.04	0.83 ± 0.00

## Data Availability

Not applicable.

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
