# Peer review of "Oxidized Dietary Oil, High in Omega-3 and Omega-6 Polyunsaturated Fatty Acids, Induces Antioxidant Responses in a Human Intestinal HT29 Cell Line"

_nutrients, 2022, doi:10.3390/nu14245341_

Round 1

Reviewer 1 Report

The article “Oxidized dietary oil, high in omega-3 and omega-6 polyunsaturated fatty acids, induces markers for antioxidant defense and ER stress in a human intestinal HT29 cell line” discusses an important topic is very well written and has excellent experimental design.

General comments

Abstract: I suggest introducing the topic before writing the objective.

Line 18: Cite some antioxidant markers.

Why did you use camelina oil and not another oil more susceptible to lipid oxidation?

Line 42-44: Citations please.

Line 48-52: Confused sentence. I suggest rewriting.

Table 1: Did you do the fatty acid composition analysis or was this information provided by the company?

Why were the times 2, 4, 6 and 9 weeks and not 2, 4, 6 and 8 weeks?

Line 547: Typo.

Author Response

Referee 1

Abstract: I suggest introducing the topic before writing the objective.

  • The abstract is rewritten. A sentence introducing the topic is included prior the objective. Line 11-25.

Line 18: Cite some antioxidant markers.

  • The antioxidant markers (GPX, ATF6, XBP1) are included. Line 19

Why did you use camelina oil and not another oil more susceptible to lipid oxidation?

  • Camelina oil was chosen because of its high content of both n-3 and n-6 PUFA which are prone to oxidation. Oxidation of camelina oil would result in an oil with different levels of oxidation products from both n-3 and n-6 PUFA, included HHE and HNE. An oil more susceptible to oxidation was not needed in this work.

Line 42-44: Citations please.

  • A reference to a review paper on the topic is added. Line 45.

Line 48-52: Confused sentence. I suggest rewriting.

  • The sentence has been changed. Line 49-53.

  • Table 1: Did you do the fatty acid composition analysis or was this information provided by the company? Yes, we did the fatty acid composition analysis by our self and the reference to the method are mentioned in line 105.

Why were the times 2, 4, 6 and 9 weeks and not 2, 4, 6 and 8 weeks?

  • In this study we wanted samples of a dietary oil with relevant levels of oxidation products from the n-3 and n-6 PUFAs, and therefore the time-points at 2,4 and 6 weeks were chosen. In addition, we wanted one samples with even more oxidation products and we chose to oxidize the oil for 9 weeks. It could of course have been for 8 weeks, but we choose 9 weeks to be sure that we got a highly oxidized sample.

Line 547: Typo.

  • The typo has been corrected. Line 557.

Reviewer 2 Report

Manuscript in very interesting, the authors proposal the increment in the antioxidant defense (response) in cells exposed to oxidized species of n-3 and n-6 PUFAs. Methodology is enough, results supported the discussion. However, I have some comments.

Major comments:

1. I suggest briefly discussing how n-3 and n-6 PUFAs regulate PPAR activity and cellular metabolic activity. PMID: 27926461

2. n-3 and n-6 PUFAs can activate the antioxidant response by regulating the activity of the transcription factor Nrf2. discuss this point.

3. It would be interesting if the authors could relate the increase in the antioxidant response with the prevention of mitochondrial dysfunction.

I Minor comments:

1. I suggest not use RE abbreviation in the title. Also, the title is some confused. Improve the title redaction

2. Improve the redaction of the aim of the study

3. To standardize the format of the figures (example: color)

4. To include a figure that showed an abstract of the principal results, and comment this figure in the discussion section

Author Response

Referee 2

I suggest briefly discussing how n-3 and n-6 PUFAs regulate PPAR activity and cellular metabolic activity. PMID: 27926461

  • The link between omega-3 and omega-6 PUFAs and PPARs is included in the discussion. Line 482-486.

n-3 and n-6 PUFAs can activate the antioxidant response by regulating the activity of the transcription factor Nrf2. discuss this point.

  • The regulation of Nrf2 by omega-3 and omega-6 PUFAs is included in the discussion. Line 465-470.

It would be interesting if the authors could relate the increase in the antioxidant response with the prevention of mitochondrial dysfunction.

  • The antioxidant response is a huge topic including mitochondrial function/integrity, and in the present study we have not investigated the effect on mitochondria specifically. We have therefore not included this topic in the manuscript.

I Minor comments:

I suggest not use RE abbreviation in the title. Also, the title is some confused. Improve the title redaction

  • The title is modified. Line 2-4.

Improve the redaction of the aim of the study

  • The aim of the study is rewritten. The subgoals are removed since the objective of the study was already described. Line 87-93.

To standardize the format of the figures (example: colour)

  • The figures from the first trial have been changed to the same colour, whereas colours for the figures of the time-study was kept in other colours to differentiate between the trials.

To include a figure that showed an abstract of the principal results, and comment this figure in the discussion section

  • The manuscript already contains a high number of figures. We don’t think that an extra figure will add substantial information to the manuscript.

Round 2

Reviewer 2 Report

Authors answered all my comments. Therefore, manuscript can be accepted in the present form.

Author Response

Thank you for your comments.